# A model for secondary traumatic stress following workplace exposure to traumatic material in analytical staff
Jessica Woodhams [1] ✉ & Fazeelat Duran [1]

Analytical professionals working in criminal justice and in social media companies are exposed to aversive details of traumatic events. Albeit indirect, exposure in these roles is repeated and can be extreme, including exposure to material containing lethal violence, sexual assault, and serious self-harm, leading to post-traumatic stress disorder and Secondary Traumatic Stress reactions. Incorporating relevant empirical research, this article considers the mechanisms that may contribute to post-exposure post-traumatic stress disorder and Secondary Traumatic Stress reactions in these roles. Building on the Ehlers and Clark model, subsequent extensions, and the authors' experience of working as, and conducting research with, law enforcement professionals, a new model is proposed to explain post-exposure post-traumatic stress disorder/Secondary Traumatic Stress reactions.

The Diagnostic and Statistical Manual of Mental Disorders Fifth Edition (DSM-5)[1] expanded the definition of post-traumatic stress disorder (PTSD) to include reactions to repeated or extreme exposure to aversive details of traumatic events. Where PTSD refers to the cognitive, emotional, and behavioural outcomes of experiencing a traumatic event first-hand (e.g., a road traffic accident or violent crime), secondary traumatic stress (STS) refers to these outcomes where the event was not experienced personally but the person is exposed to the aversive details. STS symptomatology is similar/ almost identical to PTSD symptomatology[2–4]. The DSM-5, therefore, encapsulates STS within its definition of PTSD, although STS experts concur that STS can occur without repeated or extreme exposure[4].

Significant others and professionals helping or attempting to help a trauma survivor can experience STS[5]. As such, studies of STS reflect this point and sample significant others of survivors[6] and professionals working directly with survivors; nurses, social workers, victim advocates, therapists and mental health workers, and police officers and law enforcement officials[7–9]). Not all exposure to other people's traumas is direct, and exposure for some professionals is indirect; those working with material about or depicting others' traumatic experiences (termed "traumatic material"[10], p. 112) can experience STS. Studies with personnel who work indirectly with other people's trauma (e.g., via testimonials or footage of trauma, such as child abuse investigators, those working in online child sexual abuse units, sexual violence lawyers[11,12]) have also been conducted. There are some roles within the criminal justice system and industry where exposure is entirely indirect but would certainly meet the DSM-5 PTSD criterion of repeated or extreme exposure. For example, crime analysts, intelligence analysts, and behavioural investigative advisors are exposed, in detail, to material about

the perpetration of sexual assault, murder, torture, and genocide, and that exposure often occurs daily. Social media content moderators have to engage with similar material and are described by online platforms as "the first line of defence" from this content for the rest of the online community[13] (p. 8). This material includes lethal violence, self-harm and suicide attempts, and sexual violence perpetrated against adult and child victims[13–15]. In this paper, we refer to these professionals as "analysts".

The negative impact of exposure to other people's trauma for front-line police officers and police investigators is well researched[16–18], but the same is not true for staff whose exposure is more indirect. The few qualitative studies with such employees[19–21] include accounts of symptoms of PTSD and STS[4,5]. Similarly, the negative impacts experienced by social media moderators have been reported in the popular press[22] and in a limited number of academic articles[23], but, to date, there has been no systematic, scientific research with these individuals or prevalence studies for mental disorders such as PTSD[15].

While the DSM-5[1] recognizes that secondary exposure can lead to PTSD, the exact nature of this relationship is poorly understood, and effective interventions require such mechanistic understanding. Research in this area is sorely needed but lacks a theoretical framework. While designed to explain PTSD rather than STS, we adopt Ehlers and Clark's[24] cognitive model of PTSD as a starting point in explaining how analysts' indirect workplace exposure can lead to STS/PTSD symptoms. We chose this model since it provides a detailed explanation for the development and maintenance of PTSD, and most aspects of the model are well-supported empirically[25]. We supplement this model with additional factors with explanatory value for STS/PTSD identified in the broader research

[1]School of Psychology, University of Birmingham, Birmingham, United Kingdom. ✉e-mail: J.Woodhams@bham.ac.uk

literature, thereby expanding it to incorporate further risk and resilience factors for PTSD/STS and their maintenance. Finally, we apply this expanded model to the example of analysts working with traumatic material using the limited empirical research with this group and our own experiential knowledge of conducting research with them for decades. In doing so, we identify gaps in knowledge and propose a future research agenda.

## Ehlers and Clark's (2000) cognitive model of PTSD

In Ehlers and Clark's[24] cognitive model of PTSD (Fig. 1), the individual's experience of a current sense of threat is key to explaining the symptoms that characterise the disorder (e.g., intrusions and emotional responses like anxiety and physiological arousal) and is experienced alongside them. The sense of threat can be internal and relate to the self (e.g., I am not capable, I am undeserving) and/or external (e.g., the world is a dangerous place). A current sense of threat is proposed to arise because of (1) the way it is appraised by the individual, and (2) the nature of the memory for the traumatic event, and its links with autobiographical memory. Further, the individual is proposed to engage in cognitive and behavioural strategies intended to reduce the sense of threat and/or symptoms which, paradoxically, maintain them.

## Appraisal

Ehlers and Clark[24] explain that due to the individual not appraising the threat as time-limited (i.e., occurring in the past, at a particular time), it is experienced as current and ongoing. They refer to this phenomenon as an overgeneralisation of the threat. For example, the individual may perceive all similar situations to the one associated with the trauma as dangerous. Alternatively, if the threat is internal and related to perceived personal inadequacies, then these can be appraised as long-lasting and likely to lead to similar events in the future. Victims with PTSD have more negative beliefs about self, others, and the world than victims without PTSD[25]. Appraisals of the trauma event and/or the self can lead to a range of emotional responses (e.g., shame, guilt, sadness, anger), but most relevant to PTSD is fear, which results from appraisals of threat or danger.

## The nature of the trauma memory

PTSD symptomatology, such as intrusive thoughts, is proposed to result from the individual's trauma memory being poorly elaborated and integrated within their autobiographical memory. Elaboration and integration refer to memories being stored according to themes (contextual information) and personal time periods. Without this elaboration and integration, the trauma memory is not placed accurately in time and context, adding to

the sense of the threat being current. Trauma memories that are more sensory-rich and less conceptual are more likely to be retrieved (including involuntarily) when the person encounters similar cues because their sensory richness provides numerous triggering stimuli. In support of this, victims of trauma describe their memories as sensory and perceptually rich[26].

Ehlers and Clark[24] also propose strong stimulus-stimulus (S-S) and stimulus-response (S-R) pairings for traumatic material where stimuli present at the encoding traumatic event are strongly associated without conscious awareness of this. Unawareness of these associations also makes it difficult for the individual to recognise the links between the stimulus and their trauma response which prevents them from learning that these triggers can be present without being in danger. Strong perceptual priming for such stimuli mean they are more likely to be noticed by the individual, and therefore trigger PTSD symptomatology. As a form of implicit memory, these "memory traces are not well discriminated from other memory traces" (p. 326) meaning that minimal similarity between a physical cue and the trauma event would be sufficient to trigger symptoms.

While Ehlers and Clark's[24] model proposes that contextual representations play an inhibitory role in retrieval, other research[27] demonstrates that increasing contextual representations for trauma memories facilitates their retrieval due to additional (contextual) information providing a larger number of cues for the memory. It follows that trauma memories that have been conceptually encoded could be triggered through activation of associated cues, with a greater number of cues increasing this likelihood.

Ehlers and Clark[24] also explain that the nature of the traumatic event itself can affect cognitive processing at the time of the trauma. For example, if the event was predictable, and of longer duration, then conceptual processing is more likely.

## Characteristics of the person

Characteristics of the individual are also influential; for example, prior experience of trauma can inhibit conceptual processing when encountering a new traumatic event because it reactivates trauma memories that themselves were processed in a data-driven, rather than conceptual, way. It is also recognised that individuals' life experiences change, and with these a past trauma can develop a more threatening meaning leading to a current sense of threat being experienced, or an individual may be exposed more to cues for the memory of a traumatic event. In these scenarios, there may be a delayed onset of PTSD.

**Fig. 1 | Ehlers and Clark's[24] Cognitive Model of PTSD.** Reprinted from Behaviour Research and Therapy, 38, Anke Ehlers and David M. Clark, A cognitive model of post-traumatic stress disorder, 319–345, Copyright (2000), with permission from Elsevier.

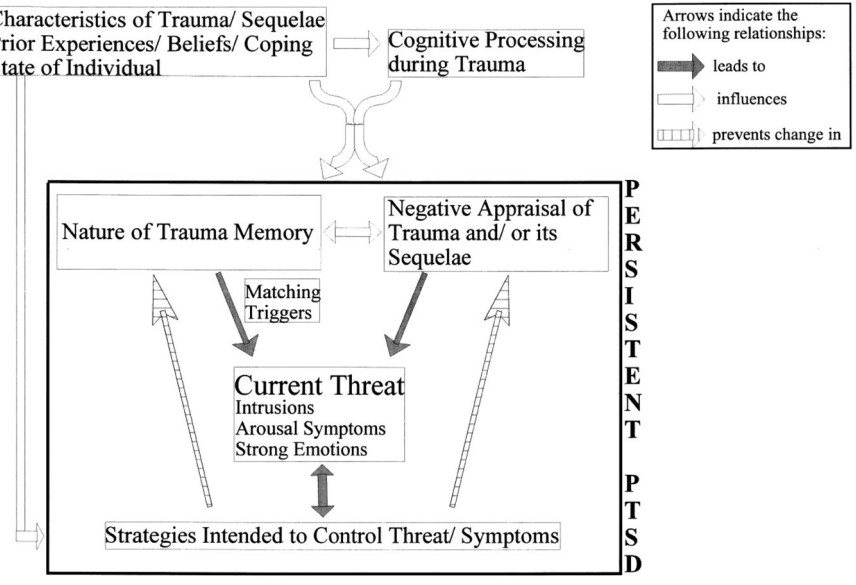

## Maintenance of PTSD via cognitive and behavioural strategies

In the model, the behavioural strategies and cognitive appraisal styles adopted by an individual can (a) directly produce PTSD symptoms, (b) prevent change in negative appraisals, and (c) prevent change in the trauma memory. For example, thought suppression, the attempt to not think about something, is argued to increase the frequency of intrusive thoughts/memories[28], and selective attention to threat cues leads to trauma-related emotional reactions and intrusive thoughts. Safety behaviours prevent the opportunity for disconfirmation of a traumatic event (fear extinction) occurring again, as does avoidance of similar situations[29]. Rumination, a cognitive processing strategy, that aims to better understand why something happened, is proposed to result in the strengthening of problematic appraisals (of self and the world[30]), which increase the current sense of threat, and stimulate emotional responses[24]. The relationship between emotion regulation difficulties and PTSD was fully explained by rumination in two cross-sectional studies[30,31], and a recent systematic review[32] cites longitudinal studies that provide further support.

## Extensions to Ehlers and Clark's[24] model

While Ehlers and Clark's[24] model can assist in understanding why professionals exposed to distressing material in the workplace develop PTSD symptoms, since its publication, other mechanisms for PTSD symptoms have been proposed. We outline how these mechanisms could be incorporated within an extension of Ehlers and Clark's model to explain why some individuals are at greater risk of developing PTSD than others.

## Mental imagery

Mental imagery "is key to bridging the experience, memory, and intrusive recollection of the traumatic event" in PTSD[33] (p. 1). Individuals vary in their propensity to use mental imagery and in its vividness[34]. Kosslyn[35] explains that mental imagery pre-trauma is associated with vivid trauma memory, and the tendency for vivid mental imagery is associated with less conceptual processing. Further, the consolidation of sensory details for traumatic events can be disrupted by engaging individuals in activities with high visuo-spatial demands soon after the trauma itself, leading to fewer subsequent intrusive memories[36]. Preliminary laboratory evidence of an association between vividness of mental imagery and intrusive memories exists[37]; however, there are no prospective studies that measure vividness of mental imagery pre-trauma exposure outside the lab. The possibility that the association between mental imagery and PTSD reflects reverse causality, therefore, remains[35].

## Social cognition

Stevens and Jovanovich's[38] (p. 1) meta-analysis concludes social cognition is an "important and understudied area" for PTSD development and maintenance. Social cognition deficits are identified in the meta-analysis as a pre-existing risk factor for PTSD development, and social cognitive ability as a resilience factor for traumatic stress, in general. Similar conclusions are drawn in Couette et al.'s[39] systematic literature review, particularly for interpersonal forms of trauma.

Mentalising, a fundamental component of social cognition, has been implicated in the maintenance of PTSD. It is the ability to understand one's mental states and those of others, and it promotes emotion regulation, a clear sense of self, reflective ability and self-control, and strong relationships with others, which can provide a sense of belonging and security[40]. When an individual is chronically fearful and hypervigilant to sources of threat[41], mentalising becomes impaired because feeling safe and secure is essential for a person to mentalise. Affective states (i.e., high arousal) can also impair the ability to mentalise at a cognitive level[40]. Social support can aid in recovery from mental illness; however, impaired mentalising weakens the ability to infer the mental states of others, and this, alongside problematic appraisals characteristic of PTSD (e.g., others cannot be trusted, I am not worthy of being helped), inhibits the individual's ability to seek and use social support[42]. It is through supportive interactions with others that the fear

response and other negative emotional states associated with the trauma are diminished/extinguished[43]. Therefore, diminished social support prevents this and can leave the individual feeling lonely and insecure, which, in turn, negatively impacts mentalising.

While these aspects of social cognition have been proposed as mechanisms for PTSD, Stevens and Jovanovich[38] conclude that exactly what components of social cognition promote either risk or resilience is unclear, prompting the need for longitudinal research to determine whether social cognitive deficits lead to a weaker social support network, or if deficits in social cognition form part of the stress response. Regarding neuroimaging studies, they note, "initial neuroimaging studies of social cognition in PTSD suggest that deficits may not be produced by alterations in the core set of brain regions involved in mentalising or emotion recognition" (e.g., the temporo-parietal junction, dorsomedial prefrontal cortex) "but may instead be related to heightened emotional responses to social signals conveying information about threat, which interfere with the ability to reason about others' internal states" (p. 3–4).

## Suppression-induced forgetting/reduced encoding

Although thought suppression at recall is proposed to be problematic in Ehlers and Clark's[24] model, others argue that suppression can stop the recollection of the fear/trauma memory through the prefrontal cortex acting on the amygdala[29] and lead to suppression-induced forgetting[44]. This direct suppression either prevents the memory coming to mind or limits its time in awareness. However, in models of memory for PTSD, which emphasise the role of data-driven vs. conscious processing in (involuntary) retrieval, the potential impact of thought suppression would be diminished when there is a strong match between the original trauma memory and the current situation, since little conscious processing would be required[29]. Thought suppression can also occur at encoding. Anderson and Hanslmayr[44] (p. 290) suggest "If, upon encoding an experience, people intentionally exclude the event from awareness, retention of the experience is impaired, compared with cases in which they intend to remember the event. Although this deficit arises from several sources, one factor is the termination of encoding by inhibition, and the disruption of episodic traces formed up until that point". Ehlers and Clark[24] cite studies that support the detrimental relationship between engaging in thought suppression at recall and PTSD symptoms. In contrast, deficits in memory suppression, diminished suppression-induced forgetting and inhibitory control have been found in participants with PTSD[45–47]. It is noteworthy that all these studies are cross-sectional and that studies of direct suppression have focused on visual images as the stimulus, yet, these are not the only type of intrusions possible.

## Emotion regulation and emotion regulation strategies

Impaired or maladaptive emotion regulation is proposed as a core component of PTSD, leading to hypervigilance and attentional biases, increased startle response, hyperarousal, generalisation of fear, and avoidance of emotional material or trauma reminders[48]. These symptoms arise from problems with both "bottom-up" appraisals of stimuli and with top-down control of emotional reactions[48], as documented in prominent theories of emotion regulation (i.e., Gross' [2015] extended process model of emotion regulation[49]); Heightened emotional reactivity, for example, contributes to a current sense of threat in those with PTSD when encountering a trigger. Resulting intense emotional arousal leads to avoidance of further triggers and the distress they cause[50]. However, doing so prevents fear extinction[24]. As previously stated, intense emotional arousal also impairs mentalising. Further, Fitzgerald et al.[48] highlight the role of impaired implicit and explicit emotion regulation (e.g., the inability to unconsciously shift attention during an emotional event vs. difficulties using cognitive strategies in a conscious manner to change an emotional response to a stimulus). It is possible that impaired implicit emotion regulation could negate the ability of an individual to engage in thought suppression at the encoding stage. Difficulties with emotion regulation are, therefore, proposed as an additional mechanism for the development and maintenance of PTSD symptomatology.

The strategies an individual uses to regulate their emotions can be adaptive (acceptance, cognitive reappraisal, and problem solving) or maladaptive (rumination and emotion suppression) when it comes to PTSD[51]. Some of these are already given explicit attention in Ehlers and Clark's[24] model, e.g., rumination. Studies adopting a prospective longitudinal design have confirmed the role of these emotion regulation strategies in persistent PTSD[51]. However, this finding has not been confirmed with longitudinal research that takes a pre-trauma measure of emotion regulation, bar Nolen-Hoeksema and Morrow's[52] findings for a relationship between rumination and PTSD. Further evidence for the role of emotion regulation strategies comes from neuroimaging studies whereby cognitive reappraisal is associated with decreased amygdala and insula activity, with the reverse observed for emotion suppression[53]. These studies underscore the importance of investigating the role of different emotion regulation strategies in PTSD and how they relate to other risk/protective factors.

## Sleep disturbance

Sleep as a behaviour is absent from Ehlers and Clark's[24] model; however, it is proposed to have a role in the development and maintenance of PTSD[54,55], particularly Rapid-Eye-Movement (REM) sleep[56]. The mechanisms for sleep disturbance leading to PTSD are via memory consolidation, emotional memory processing, cognitive control, and emotion regulation[54,55,57–59]. Sleep disturbance impacts elements of socio-emotional functioning (e.g., recognition of emotional states from faces, emotional expressivity, emotional intelligence[60]), and some research shows a direct relationship between sleep disturbance and social cognition and social competence, albeit in individuals with other mental illnesses[61,62].

Hyperarousal associated with PTSD can also make it difficult for an individual to fall asleep[57]. As well as poor sleep being a consequence of emotional reactivity, it is proposed as a causal factor. For example, a genetic predisposition for dysfunction in sleep-wake regulating neural circuitries can, alongside stressors, lead to poor sleep and negative affect. Furthermore, sleep deprivation impacts the prefrontal cortex's (PFC) ability to modulate emotional reactivity[57]. In addition to the prefrontal cortex, the amygdala, insula, and hippocampus are sensitive to sleep disruption[63], and all are implicated in the development and maintenance of PTSD.

Few prospective longitudinal studies exist for sleep disturbance and PTSD, and even fewer have a pre-trauma measurement of sleep disturbance, which greatly limits our understanding of their temporal association. Gehrman et al.'s[64] study of veterans did include a pre-trauma assessment prior to military deployment; however, only 50% of the sample had experienced combat-related trauma during deployment. Van Liempt et al.[65] also conducted a pre-deployment measure of sleep quality with veterans, and all their participants were exposed to war-time stressors. However, for both studies, the assessment of sleep quality was limited (i.e., hours of sleep and reporting insomnia symptoms for Gehrman et al., or reported insomnia symptoms and nightmares for van Liempt et al.), and REM sleep was not measured. Studies could be improved through the adoption of objective measures of sleep, such as actigraphy[57].

## An expanded model of PTSD/STS for analytical staff

Worldwide, a significant number of employees are exposed regularly to traumatic material in the workplace (such as in criminal justice settings and social media companies). Exact figures are hard to obtain; however, the Internet Crimes Against Children (ICAC) task force employs thousands across the United States[7] and, in 2014, it was estimated that 100,000 people were employed as content moderators worldwide (Chen, 2014, as cited in ref. [15]). Prevalence figures for PTSD/STS in professionals exposed to other people's trauma show a sizable minority are affected: the cumulative prevalence rate for PTSD in an eight-month period for Canadian lawyers was 13%[12], and 25% of the ICAC task force reported high levels of STS[11]. Both roles can involve some direct work with trauma survivors/perpetrators alongside indirect work. Levin et al.[66] considered the directness of exposure to other's trauma by categorising their sample of forensic science professionals into field vs. lab-based. Those field-based reported higher levels of STS than those lab-based, although both showed moderate levels of STS. However, some lab-based professionals' work would have "personalised" the crime, and others not. We found no other study that measured STS prevalence in professionals working indirectly with other people's trauma (similarly, Steiger et al.[15] report no prevalence studies for content moderators). However, qualitative research with analysts whose work-based exposure is entirely indirect (and very "personalised") details symptoms consistent with STS[20,21]. Lavis'[20] work with police and law enforcement analysts reported precautionary behaviour (e.g., locking windows, carrying keys in hand), difficulty in making new friends due to lack of trust, hypervigilance to threat cues, physical and mental exhaustion, and intrusive thoughts. Other police analytical staff reported nightmares of the cases they were working on and strong negative emotions like anger, irritability, sadness, crying, or disgust when exposed to the details of serious crimes[19]. Digital forensics analysts described the negative effects of their work on them as long-term/permanent and irreversible[21].

Given the number of employees working in roles with (daily) exposure to traumatic material and moderate/high levels of STS and PTSD symptoms in associated samples, research with these populations is sorely needed and so too is a guiding theoretical framework. We, therefore, consider how the mechanisms outlined in Ehlers and Clark[24], and the additional mechanisms we have identified, relate to the work of staff in these roles in an attempt to explain how workplace exposure to traumatic material could lead to STS/PTSD symptomatology (see Fig. 2 for our model). While we refer to PTSD/STS in our model, the factors we have identified are likely to also lead to depression, anxiety, and sleep disturbance, since there is often co-occurrence between PTSD, depression, and anxiety[67,68]. While some longitudinal research has found PTSD to lead to anxiety and depression (but not vice versa)[69], other studies report bidirectional relationships between PTSD, depression, and insomnia[68,70].

## Appraisal

In the first author's 18 months as a crime analyst, she was exposed to the details of hundreds of sexual offences. Dosage, in various forms (e.g., caseload volume, ratio, and frequency of exposure) is a significant risk factor for STS for professionals working with trauma victims[3], for attorneys[71], and for professionals indirectly exposed to the details of crimes against children[11,72]. There is also expert consensus on it being a likely risk factor for STS in general[4]. We propose that analysts form a memory for each case they work on, therefore, high dosage will lead to a large number of memories of other people's traumatic experiences. Moreover, high dosage would result in appraisals that the world is a dangerous place due to the sheer volume of traumatic events to which they are exposed. We also propose that high dosage brings with it many potential triggers for memories of other people's traumas. We would, therefore, expect a longitudinal study to show a positive relationship between dosage and PTSD/STS symptomatology.

In addition to dosage in general, exposure also brings new knowledge that traumatic events occur in different contexts and to people of all characteristics. For example, child sexual abuse can be perpetrated in the home, at school, by peers and by adults, and by perpetrators known and not known to the victim. Similarly, stranger sexual offences occur at indoor locations (home, work, shopping centres, car-parks, on public transport or in taxis, at bars and nightclubs) and at outdoor locations (public parks, on the street, at the side of a river or canal), throughout the year, at differing times of the day, and to all genders, ethnicities, and age groups. Analytical professionals are, therefore, likely to appraise many situations as potentially dangerous: digital forensics analysts classifying child sexual abuse material reported that child sexual abuse and the perpetrators are everywhere and that "there is nowhere that is actually completely safe"[21] (p. 04). Analytical staff also report negative appraisals of others: some report perceiving all men as potential sexual predators[19], or other parents as potential child sexual offenders[20], or they report they cannot trust anyone[21].

As per Ehlers and Clark[24], negative appraisals of the world and others can produce a current, and generalised, sense of threat and emotional reactions such as fear (for appraisals of threat), anger (for appraisals of

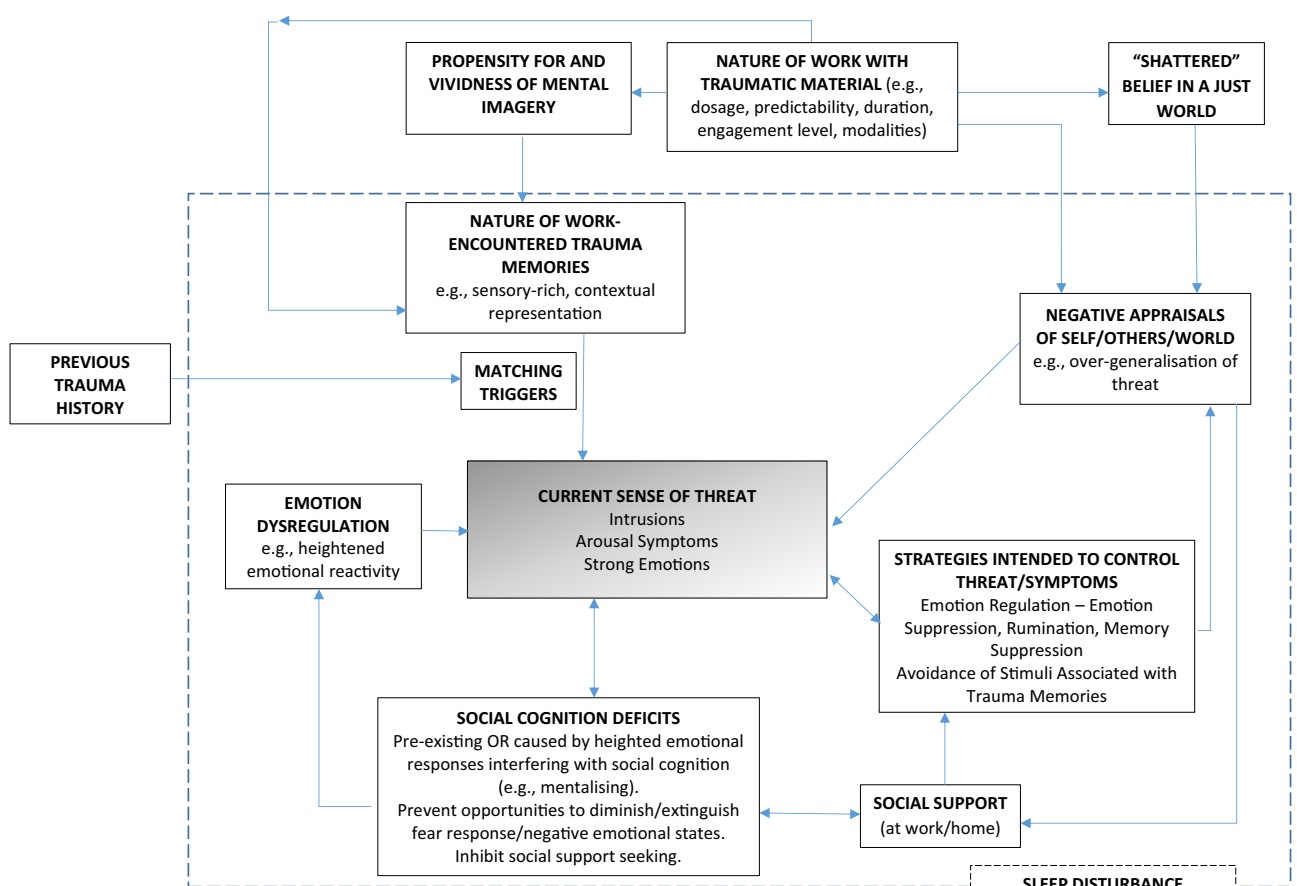

**Fig. 2 |** A model of the mechanisms via which workplace exposure to traumatic material could lead to STS/PTSD-type symptoms. The impact of sleep on many of the mechanisms within the original and the extended model, as referred to above, is represented by a dashed box in the diagram where empirical research has demonstrated that everything within the box has the potential to be impacted by sleep quality.

unfairness), and sadness (for appraisals of loss). Analysts report all of these reactions following exposure to traumatic material[19]. We propose that workplace exposure characterised by traumatic events occurring in a broad range of contexts could have a negative effect in addition to that stemming from dosage alone. An analyst experiencing hypervigilance for threats and chronic fear would have difficulty in mentalising, further contributing to the maintenance of PTSD symptoms.

## Nature of trauma memory
Equivalent research to that with trauma survivors, which documents sensory-rich memories[26], does not exist for analytical professionals. However, they do report intrusive memories of, and flashbacks to, traumatic material[19,20], which suggests that similar mechanisms could explain STS as those that feature in theories/models for PTSD for personally experienced trauma. This line of argument leads us to consider what characterizes workplace exposure to traumatic material and how the nature of workplace exposure might influence the formation of memories of traumatic material.

Analysts working with material regarding someone else's trauma have not personally experienced it, which could impede its accurate placement in time within autobiographical memory. This lack of accurate integration would increase its likelihood of being triggered, particularly when encountering similar cues[24]. For example, the poorly integrated memory of a sexual assault that occurred in a nightclub could be triggered for an analyst when presented with cues for nightclubs (such as when attending one for a social event). On such occasions, Ehlers and Clark's[24] model would predict that the analyst is at risk of experiencing intrusive thoughts or flashbacks to the traumatic material and that poor integration into autobiographical memory adds to the sense of current threat.

Analysts receive information about the traumatic experiences of other people via different mediums. For example, some professionals categorise child sexual abuse material, view photographs of victims of genocide or war, or read the written accounts or transcripts of videos of witnesses/victims. They are, therefore, receiving information via visual means. Professionals listening to accounts of abuse or other violence are receiving information via solely auditory means. Watching a victim give their testimony or watching a video recording (with sound) of a traumatic event involves both modalities. Given the increase in sensory information available in the formation of memory in the latter scenario, we propose it is a higher-risk scenario for developing PTSD/STS. Case-to-case variation is also likely for traumatic material in terms of the details provided. For example, two video-recorded victim/witness testimonials could vary in their level of detail due to the interviewer's competence and/or the survivor's ability to recall and articulate their experience[73]. Variation in detail could affect the amount of contextual information provided about a traumatic event, its encoding, and subsequent involuntary recall, as per Pearson et al.[27], and/or it could lead to a larger amount of sensory information in one testimonial compared to another, relevant to the mechanisms proposed by Ehlers and Clark[24].

Longer duration of exposure to traumatic material could also increase the amount of information (conceptual or sensory) encoded into memory. Variation in duration of exposure could occur due to the nature of the analyst's task requiring short vs. lengthy engagement with material, as well as the actual length of some types of traumatic material (e.g., differing lengths of video footage). Where an analyst's role requires them to repeatedly re-visit the material (e.g., in crime linkage analysis[74]), this requirement would also increase duration of exposure. The amount of time spent exposed to traumatic material is something that some employers actively

monitor and limit to protect their staff[75], and analysts report taking self-imposed breaks[20].

The degree to which an analyst engages with traumatic material in performing their role could affect its encoding into memory. Digital forensics analysts engage with child sexual abuse material to (1) determine if the person depicted in the image/footage is a child and (2) they must categorise the image according to the level of indecency depicted[76]. Crime analysts engaged in crime linkage analysis determine if two or more crimes are linked to one another, which involves recording the behaviour displayed by the perpetrator, interpreting it, and comparing it to behaviour in other offences[74]. The source material for crime linkage analysis of sexual offences is the narrative of the crime provided by the victim via a written statement, a transcript of a police interview with the victim, or the actual video recording of the interview. Both types of analytical work involve engagement with the material to arrive at a decision; however, we would propose the latter type of analysis requires a greater level of engagement and conceptual processing. Ehlers and Clark's[24] model would predict that such engagement would be protective, reducing the likelihood of later intrusions. However, Pearson et al.'s[27] findings suggest the opposite, whereby greater engagement leads to more information about the traumatic event being encoded, producing more cues and more associations between them.

In Ehlers and Clark's[24] model, the predictability of encountering a trauma increases the likelihood of conceptual processing, which would reduce the prospect of later intrusions. In some analytical units, a form of triage or review of a case is conducted by one member of staff before it is handed over to the member of staff who will do the in-depth analysis[77]. Working practices like this provide opportunities to prepare colleagues regarding the content of the material and analytical professionals in criminal justice settings value being prepared in this way[19]. However, it should also be remembered that even in workplaces where exposure to traumatic material is expected, exposure can still occur unexpectedly (e.g., an image could flash up on a colleague's screen in a group office).

## Mental imagery
Mental imagery can form from short-term or long-term memories[78]. Visual mental imagery can form in short-term memory when an analyst views an image of a crime scene or a victim, or footage of a violent act, and creates an image in their mind using the perceptual information available. However, analysts may also create mental imagery by intentionally or unintentionally drawing on visual representations in their long-term memory relevant to the traumatic material with which they are working. In doing so, they associate mental imagery that is personally relevant to the memory being created for another person's traumatic experience. It would follow, from Pearson et al.'s[27] research, that this association would increase the likelihood of memories of the traumatic material being recalled later via the creation of associations between the trauma memory and the analyst's episodic memory. Individuals with a greater propensity for (vivid) mental imagery may also store trauma memories characterised by greater perceptual or sensory detail. According to Ehlers and Clark[24], such sensory-rich memories would be more easily triggered, leading to PTSD symptomatology. Since individuals differ in mental imagery ability as well as vividness[78], the extent to which memories are triggered will likely vary between individuals.

## Maintenance of PTSD/STS via cognitive and behavioural strategies
Impaired or maladaptive emotion regulation is proposed to result in hypervigilance, hyperarousal, and avoidance of trauma reminders, as well as heightened emotional reactivity in general[48]. Qualitative research with analysts working with traumatic material has documented all these outcomes following exposure[19,20]. However, no quantitative studies investigating the relationship between PTSD/STS and emotion regulation have been published for analysts specifically. In qualitative studies, they report engaging in a range of emotion regulation strategies: detachment, emotional distancing, reinterpretation, cognitive reappraisal, and selective attention[19,20]. Several strategies hypothesised to be maladaptive have been

reported by analysts: thought suppression of involuntary recall of memories of traumatic material, avoiding similar situations to those documented in traumatic material, and safety behaviours associated with events learned about through exposure to traumatic material[19].

Thought suppression at encoding is considered an adaptive strategy[79], and while analysts cannot exclude a traumatic event from awareness entirely, since their role is to analyse these events, they do report coping by engaging with the material in a limited way (e.g., focusing on the facts of the case and/or not engaging with the emotional content[19,20]). The latter might also lead to fewer contextual representations being formed, reducing the likelihood of involuntary retrieval[27]. When analysing traumatic material (and therefore encoding information into memory), analysts also report engaging in dissociation (i.e., derealisation and emotional numbing/suppression). In Ozer et al.'s[80] meta-analysis of predictors of PTSD, peri-traumatic psychological processes had the largest effects on rates/levels of PTSD symptoms. In particular, they considered self-reported peri-traumatic emotional response and self-reported peri-traumatic dissociation. The former refers to the intensity of negative affect felt at the time of the traumatic event, and the latter to depersonalisation, derealisation, and emotional numbing. While this has yet to be studied quantitatively in the context of STS, it is noteworthy that in interviews, some analysts report intense emotional responses to the traumatic material (e.g., breaking down in tears, intense anger and disgust[19,21]), as well as using derealisation and emotional numbing (i.e., emotional suppression) to distance themselves and mute their emotional response. Whether these translate in the same way as risk factors is a question for investigation. In experimental studies where participants have viewed aversive imagery, self-reported dissociation was associated with intrusive memories[81]. It is very difficult to assess peri-traumatic emotional response and dissociation with personally experienced traumas except by using retrospective self-report[80], and in the lab, there are ethical issues with exposing people to extreme traumatic material to which they would not normally be exposed[81]. However, with their consent, it would be possible to assess these two factors prospectively with analytical staff exposed to distressing content as part of their work using measures of physiological arousal and brain imaging to overcome the limitations of self-report.

## Sleep disturbance
As per our amended model (Fig. 2), poor sleep is implicated in both the development and maintenance of PTSD via its potential impact on several of the risk factors for PTSD/STS. With regard to working with traumatic material, analysts from several different countries report sleep disturbance[19]; however, its exact role in PTSD/STS remains to be established.

As proposed by Ehlers and Clark's[24] model, we also consider here how the characteristics of the individual could increase or decrease the likelihood of PTSD/STS symptomatology in these roles. The potential influence of the nature of the work with traumatic material has already been discussed above. However, we also consider here how the workplace environment in which exposure occurs could be influential.

## Characteristics of the person
Individual differences exist for several potential risk factors that have been discussed already: mentalising and other forms of social cognition[82], inhibitory control[79], which is relevant to suppression ability at encoding and recall, and vividness and use of mental imagery[78]. For people with high vividness of mental imagery, repeated presentation of threatening stimuli can lead to sensitisation of fight/flight reactions[35], therefore dosage, or the requirement to re-visit the same traumatic material repeatedly, might be particularly relevant in analysts who experience vivid mental imagery. In accordance with Ehlers and Clark[24], analytical staff who have experienced traumatic events themselves would be hypothesised to be at greater risk, particularly if their personal traumas are similar in nature to traumatic material they are exposed to at work. A meta-analysis by Hensel et al.[3] found personal trauma history to be a significant risk factor for STS (with a small effect size) for professionals working directly with trauma victims, and a

**Table 1 | Variables for future research and their predicted relationship with risk for PTSD/STS symptomatology**

| Individual variables | Work-related variables |
|---|---|
| **Propensity to use and vividness of mental imagery ↑** | **Dosage of TM ↑** |
| **Unresolved personal history of similar trauma ↑** | **Working with the TM via dual modality ↑** |
| **Tendency to ruminate ↑** | **Exposure to TM about events occurring in a range of contexts and to differing people ↑** |
| **Belief in a just world ↑** | |
| **Sleep disturbance (e.g., reduced REM sleep) ↑** | **Personal relevance of the TM ↑** |
| **Use of emotion suppression ↑** | **Working with detailed TM ↑** |
| *Ability to mentalise ↓* | **Longer duration working with TM/re-visiting the same TM multiple times ↑** |
| *Inhibitory control ↓* | |
| *Use of cognitive reappraisal ↓* | **Engaging with the TM in greater depth ↑** |
| *Use of acceptance ↓* | **Workplace stress (separate from exposure to TM) ↑** |
| *Use of problem solving ↓* | *Predictable exposure to TM and opportunity to prepare ↓* |
| | *Social support (at work and/or home) ↓* |
| | *Workplace culture that promotes mentalising ↓* |

TM traumatic material, ↑**hypothesised increased risk**; ↓ *hypothesised decreased risk*.

systematic review by Leung et al.[8] found the same relationship for mental health workers also working directly with trauma victims. However, Sprang et al.[4] caution against assuming that a personal history of trauma will always act as a risk factor and argue that if PTSD associated with previous traumas has been resolved, the risk of subsequent STS is reduced. As per Ehlers and Clark's[24] model, with changing life circumstances, an analyst's personal life (e.g., becoming a parent) could lead to traumatic material developing a more threatening meaning.

In Ehlers and Clark's[24] model of PTSD, appraisals lead to emotional responses. While it does not feature in their model, we propose that Belief in a Just World[83] is worthy of future study as a potential risk factor. This concept refers to an individual's belief that the world is a fair and just place, and analysts certainly report protecting others and achieving justice as motivators for staying in their roles[19–21]. Ongoing exposure to details of traumatic events, such as crimes or harmful online behaviour, would directly challenge belief in a just world[19]. Drawing on Bolton and Hill[84], we propose that having your belief in a just world challenged daily would likely lead to appraisals of unfairness, unpredictability, helplessness, and, ultimately, psychological distress. Appraisals of unfairness are specifically implicated in Ehlers and Clark's[24] model of PTSD for their role in emotional responses, particularly anger. Further, Bai et al.[85] found repeated trauma "shattered" belief in a just world (i.e., shattered assumptions theory[86]) leading to STS/PTSD symptoms (e.g., hyperarousal). We, therefore, hypothesise that newly recruited analysts with a strong belief in a just world are at greater risk of PTSD/STS symptoms following exposure to traumatic material than those with a weaker belief in a just world.

## Characteristics of the workplace
The workplace can act as a facilitator, or an inhibitor, of mentalising. Stress, in general, inhibits mentalising[40] and significant workplace stress associated with pressure to meet targets has been reported in three qualitative studies with analysts[19–21]. Time pressure also prevents them seeking workplace support[19].

As reported, impaired mentalising from PTSD inhibits a person's ability to seek and use social support. Social support is a protective factor in models of STS[10], and workplace support, specifically, was a protective factor for STS in Hensel et al.'s[3] meta-analysis. Supervisory support was significantly associated with less STS in professionals working on internet child sexual abuse[72]. Similarly, analysts working with traumatic material report

support from supervisors as protective[19,21]. However, a positive relationship with a supervisor does not guarantee support-seeking from a supervisor due to concerns about being redeployed[21]. Support from colleagues is also key to coping with workplace exposure to traumatic material[19,87]. Sources of social support in the workplace are particularly important where confidentiality limits discussion of work with friends and family[19].

A work environment that encourages mentalising and/or is STS-informed is, therefore, more likely to succeed in protecting its employees[41]. Although not designed to capture mentalising, interviews with analytical staff working with traumatic material give examples of how the workplace can support mentalising (e.g., having a safe space to reflect on material and the cognitions and emotional responses it elicited[86]).

## Outlook
Based on our elaboration of Ehlers and Clark's[24] model, we have identified a list of variables that we predict will either increase or decrease the risk of PTSD/STS symptomatology in analysts (Table 1). In articulating these, we intend to provide an impetus for further research with these professionals. For all variables identified, a prospective longitudinal study with pre-exposure measurement of the variable of interest is key. Since the existing evidence for thought suppression at recall is mixed, we have not specified a direction for the relationship, and it is not placed within Fig. 2. Similarly, for suppression at encoding, it is unclear at this point how successful attempts to suppress will be given the nature of an analytical role is to engage with the traumatic material in a conceptual way. While dissociation during a traumatic event has been noted as a risk factor for PTSD, it is not clear at present whether analysts adopting dissociation strategies when working with traumatic material leads to a similar outcome, but this possibility should be investigated. We also expect that there will be interactions for some variables: for example, propensity to use and vividness of mental imagery and the nature of the analyst's role may interact, whereby the nature of the role might moderate the relationship between mental imagery and PTSD/STS symptoms. Furthermore, the relationship between belief in a just world and PTSD/STS symptoms might be mediated by dosage. Discovery of the nature of the relationships between these variables and PTSD/STS symptomatology is needed to enable the development of evidence-based interventions for prevention or treatment.

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

## Acknowledgements
The authors would like to thank Professor Andy Bagshaw and Dr Stephane De Brito for their suggestions on an initial draft of the paper.

## Author contribution
JW was responsible for the conceptualisation of the paper, for reviewing the literature, and writing the first draft of the paper. FD contributed to the conceptualisation and provided edits to the paper.

## Competing interests
The authors declare no competing interests.
