## [Peer Review File · Communications Psychology]

17th Apr 23

Dear Jess,

Thank you for your patience during the peer-review process. Your manuscript titled "A Model for the Impact of Work-Place Exposure to Traumatic Material in Analytical Staff" has now been seen by 2 reviewers, and I include their comments at the end of this message.

Given the long wait, I'm all the more pleased to let you know that the expert referees are in principle enthusiastic about the idea and subject of your review. However, they also mention a number of concerns which can only be addressed through extensive revisions. We are interested in the possibility of publishing your manuscript in *Communications Psychology*, but would like to consider your response to these concerns in the form of a comprehensively revised manuscript before we make a decision on publication.

As you can see from the reports, the referees' primary concerns are presentational, but they also raise a list of critical points regarding individual statements that need to be better argued and more compellingly supported by the literature.

We recommend strongly that you build on Reviewer #1's proposal for an alternative structure of the piece, which we agree will make it easier for readers to follow and critically evaluate your argument. Please also note Reviewer #2's constructive comments with regard to reordering how individual arguments are presented in the piece, and the opportunity to create a stronger focus and common theme throughout the manuscript. Although your text essentially conveys a proposal, that does not mean that this novel aspect should be presented last. Ideally, the new framework would be presented after the reader has been familiarized with the reference model, and the remainder of the text would investigate the what evidence supports or fails to support the new model, or the hypotheses derived from it. An outlook section is valuable, and as the reviewers suggest, may focus either on necessary research or on suitably caveated implications for interventions (or both).

In sum, we invite you to revise your manuscript taking into account all reviewer and editor comments.

EDITORIAL POLICIES AND FORMATTING

You will find a complete list of formatting requirements following this link:

<https://www.nature.com/documents/commsj-style-formatting-checklist-review-perspective.pdf>

Please use the checklist to prepare your manuscript for resubmission.

* **TRANSPARENT PEER REVIEW:** *Communications Psychology* uses a transparent peer review system. This means that we publish the editorial decision letters including Reviewers' comments to the authors and the author rebuttal letters online as a supplementary peer review file. We publish these records for all accepted manuscripts. However, on author request, confidential information and data can be removed from the published reviewer reports and rebuttal letters prior to publication. If your manuscript has been previously reviewed at another journal, those Reviewers' comments would not form part of the published peer review file.

If you have any questions about any of our policies or formatting, please don't hesitate to contact me.

Please use the following link to submit your revised manuscript and a point-by-point response to the referees' comments (which should be in a separate document to any cover letter):

[link redacted]

We hope to receive your revised paper within 12 weeks; please let us know if you aren't able to submit it within this time so that we can discuss how best to proceed. If we don't hear from you, and the revision process takes significantly longer, we may close your file.

We understand that due to the current global situation, the time required for revision may be longer than usual. We would appreciate it if you could keep us informed about an estimated timescale for resubmission, to facilitate our planning. Of course, if you are unable to estimate, we are happy to accommodate necessary extensions nevertheless.

Please do not hesitate to contact me if you have any questions or would like to discuss these revisions further. We look forward to seeing the revised manuscript and thank you for the opportunity to review your work.

Best wishes,
Marike

Marike Schiffer, PhD
Chief Editor
Communications Psychology

REVIEWERS' EXPERTISE:

The reviewers have extensive expertise in trauma research, including research on workplace and vicarious trauma.

REVIEWERS' COMMENTS:

Reviewer #1 (Remarks to the Author):

The reviewed manuscript is a theoretical paper that initially argues for the relevance of the Ehlers and Clark cognitive model of PTSD as applied to professionals exposed to traumatic content and material through their analytical work, potentially leading to the development of secondary trauma. The study of analytical professionals in this context is indeed lacking, as most of the research and clinical literature is focused on those directly experiencing a personal traumatic event. Thus, there is value in the position being raised by the authors for overlaying how the Ehlers and Clark model could

explain the development and persistence of secondary trauma reactions for this vulnerable employee group. However, I have several recommendations intended to enhance this manuscript (see below), as well as a general recommendation to clarify the goals of the current paper. Yes, the application to analysts comes through, but the additional elements to this model go beyond analysts and speak to general components that could be added to the model regardless of whether the individual is an analyst or not. Thus, I wonder if the authors should re-orient the paper to first describe the original Ehlers and Clark model, speak to how this model applies to secondary trauma in general first, and then add their expansions to this model, and then make the application specific to analysts as it might apply in the context of secondary trauma.

Note that page numbers are missing from the document, so I will use line numbers to identify specific areas of reference in my comments below.

1. Line 72 - the authors first introduce the term secondary traumatic stress on page 2 of the introduction. Given that this paper hinges on this conceptual application, I would like to see it operationally defined in more detail with stronger links to the existing research on the nature of this traumatic stress exposure and the clinical presentation prior to giving into the specific application of the Ehlers and Clark model to analytic professionals.
2. Line 93 - clarify the first sentence of the appraisal paragraph. An event can be time-limited and still be perceived as threatening or potentially harmful; thus, how are Ehlers and Clark defining time-limited in this context?
3. Line 135 and throughout the manuscript - Rather than use the term "rape", I recommend that the authors use the term "sexual assault" which can include a broader range of invasive non-consensual sexual behaviours. In addition, the statement that "rape" is relatively frequent must be supported by a research citation or source. In fact, sexual assault is one of the lower-reported forms of violent crime when these statistics are reviewed, including when viewing official crime reports and victim survey reports. Thus, this statement needs to be qualified as to what is meant by frequent and in reference to what benchmark.
4. Line 139 and paragraph - I encourage the authors to consider applying the theory to the analysts own memory of what they are reading/viewing in their work, rather than focusing on the memory report of a third party as the primary stimuli source. Yes, the third party's account of their traumatic experience may influence the level of secondary trauma exposure and the subsequent impact on the analyst, but the analyst's own memory and interpretation of that content is likely key to the subsequent psychological impact that may lead to trauma reactions over time.
5. Line 160 - report the known validity of the SAM and VAM as mechanisms, and acknowledge the criticisms of these components in the PTSD experience.
6. Where does mentalizing fit within the expanded model? Is it a vulnerability factor in how memory works when exposed to secondary trauma that influences encoding and memory retrieval - or is it in the appraisal per se or a sensory experience issue? I also do not see it represented as a factor in Figure 1.
7. When discussing the role of suppression, clarity is required to separate out the effects (or theoretical effects) of suppression during the event itself (or at initial exposure to the problematic

stimuli) versus post-event suppression, which relates to recall effects rather than how an event was encoded initially. Presenting the literature to discuss these two phases of when suppression may occur would be helpful in clarifying the findings in the literature and understanding the influence on trauma reactions and persistence over time.

8. line 337 - "unpick" should be "unpack"

9. instead of ending the paper with a list of hypotheses for testing in subsequent research (I would move those to a table), the authors should conclude with potential intervention or prevention strategies that would be helpful for analysts to use based on their theoretical arguments - what is the value of this theoretical argument and expansion of the Ehlers and Clark model to analysts? That should be the take-home message of the paper.

Reviewer #2 (Remarks to the Author):

The paper departed from the cognitive model of Ehlers and Clark and aims at formulating a set of hypotheses by expanding this model to workplace exposure. The paper considers the mechanisms for post-trauma psychological distress, particularly PTSD-type reactions.

1. While there is great interest in the topic there are several shortcomings in the paper in its current form that limit potential for publication. To start with the end: the hypotheses that are provided at the end of the paper, to aid at providing a theoretical framework, lack cohesiveness; the arguments are still loosely connected while throughout the paper scattered references are provided. The strong examples in the brief introduction of the paper are missing in the various paragraphs in the paper and fail to help better understand what the drivers for the distress or disorder-like symptoms are. Several questions remain: e.g. what is traumatic material, how does the frequency or dose of exposure count, is there a measure for the extreme traumatic material? Is there information on the duration of the work exposures, in terms of days working? These just some of the variables that also need to be taken into account to expand the cognitive model.
2. The first line of the outlook paragraph states the central theme of the paper. There should be more focus on this throughout the paper, in terms of evidence, examples, and evidence of the impact, it now fails to give enough leads as to how this can be studied. Based on these limitations, the hypotheses are raising more questions.
3. As authors state repeatedly that longitudinal studies are needed to unpick issues, they do not specify what these studies need to do, answer or have in addition then being longitudinal. If authors make a statement that 'there are few prospective longitudinal studies and...', they perhaps should have made this statement based on a systematic review, as of now the statement misses its foundation.

Some other points of concern:

- The title is very interesting. Workplace exposure to traumatic material can be seen in a wide variety of situations. And has been introduced in DSM5 with a lot of discussions. The introduction gives ample situations in which this is presented. Yet, the definition of what traumatic is in the material needs to be understood before the impact and the context in which this is occurring can be reviewed.
- The paper is presented as a review in line 83, but it is not clear what kind of review this paper is providing.

- The authors should do better and be talking about 'potential' impact when it comes to trauma. It is not so that the impact is similar for everyone.
- The job demands bring on exposure to traumatic material. A discussion would be relevant if it matters that the exposure is anticipated, and not a surprise. The exposure/analysis of elements of the material is part of the assignment.
- When stated as such, why is research in the area sorely needed, please motivate once more. The motivation should be based on empirical data about symptom manifestations leading to the inability to perform the job (criterion F, G).
- Post-traumatic distress is not post-traumatic stress disorder.
- What is the role of circumstantial factors, if someone just becomes a father, or has a death in the family, would they not be more vulnerable to the distress potentially leading to disorder-like experiences/symptoms. Is leadership not aware of this and removes the person from their job temporarily? It is unlikely that 'one' exposure leads to PTSD. It is (likely) repeated exposure. So the paper needs also to mention the (repeated) pressure in job situation. It would be interesting to see what impact is of measures to take distance or request (temporary) replacement.
- The paragraph starting with a review of mechanisms is valid; appraisal, is context dependent; threat is ok, unfairness is not well worked out. In some, it may be a motivation to pick a job just in working to fight against the unfairness.
- The prevalence of rapes and other traumatic events is not a valid precedent for claiming that exposure to it is traumatic.
- Memory paragraph, here the argument is weak. Why emphasize rape and stimulus generalization, when other forms of trauma are also prevalent and may cause more workplace distress or trauma. E.g. forensics in police and crime cases. Here the examples given in the introduction could be more worked out to make a better case.
- Senses matter – also weak since a lot of material is not only visual and auditory but also reading, or in movie-like presentation. It is to be demonstrated if the reference can be extrapolated. The reference is on learning, wonder if the statement is well referenced with this paper, which is on learning.
- Some reference could be better chosen, e.g. when it comes to early life trauma.
- Extending Ehlers and Clark to Mental imagery – and suppression at encoding and emotion regulation and social cognition. It is to be regretted that they do not carry the focus of the paper on workplace.
- Sleep is a causal and maintaining factor – this is an example of unclear information. The paragraph lacks focus. Is not linked to the topics of the paper. Half a page is on sleep that is unrelated to the title.
- Social cognition only has a few lines on social support and support seems very important. How is this weighted in relation to social cognition.
- The figure is interesting but lacks the finetuning that is needed.
- Line 335 TPJ and dmPFC , spell out.

Response to Reviewers – Communications Psychology - COMMSPSYCHOL-23-0010

Our sincere thanks to the reviewers for the time and effort spent reviewing our manuscript. The suggestions have been incredibly helpful and we explain below how we have addressed them (third column). In engaging in wider reading, we have uncovered other relevant articles and mechanisms therefore these have been incorporated into the manuscript despite this not being a request. The paper has been reorientated and first introduces (briefly) the need for a theoretical framework for understanding STS particularly with groups *indirectly* exposed to other’s trauma. It goes on to describe Ehlers and Clark’s (2000) model, now in more detail. The next section considers how this model could be extended to include other mechanisms for PTSD/STS, and, finally, we focus specifically on analytical professionals, giving more details regarding the nature of their work. This led to a much longer (but clearer – thank you) manuscript therefore we have done our best to also cut out words. We have amended our extended model (Figure 2) accordingly. Unfortunately, we weren’t able to highlight in red what we have changed because the paper has had a substantial re-write so we’ve tried to indicate relevant page numbers in the table below.

R1	1. Line 72 - the authors first introduce the term secondary traumatic stress on page 2 of the introduction. Given that this paper hinges on this conceptual application, I would like to see it operationally defined in more detail with stronger links to the existing research on the nature of this traumatic stress exposure and the clinical presentation prior to getting into the specific application of the Ehlers and Clark model to analytic professionals.	STS is now clearly related to PTSD on p. 3 of the manuscript and the relationship explained. The overlap in symptomatology is clearly stated with appropriate references on p.3. Where PTSD and STS do not overlap is covered in a sentence with an appropriate citation (p. 3). Prior research on STS is included in the paper with reference to groups working directly with trauma survivors and others working more indirectly. This includes statistics on prevalence of PTSD and STS in these groups and relationships between STS and relevant risk factors (see p. 15, 16 and 17).
R1	2. Line 93 - clarify the first sentence of the appraisal paragraph. An event can be time-limited and still be perceived as threatening or potentially harmful; thus, how are Ehlers and Clark defining time-limited in this context?	We have returned to the paper to ensure we are representing Ehlers and Clark correctly and have added an explanation of what they mean. See p. 6.
R1	3. Line 135 and throughout the manuscript - Rather than use the term "rape", I recommend that the authors use the term "sexual assault" which can include a broader range of invasive non-consensual sexual behaviours. In addition, the statement that "rape" is relatively frequent must be supported by a research citation or source. In fact, sexual assault is one of the lower-reported forms of violent crime when these statistics are reviewed, including when viewing official crime reports and victim survey reports. Thus, this statement needs to be qualified as to	We have changed rape to sexual assault throughout. We recognise that our explanation for why analysts might perceive rarer crime types (i.e., rape) wasn’t very clear. We have now talked about this in the context of dosage and also explained in much more detail what we mean by analysts encountering a broad range of situations for such crimes, leading to, in our opinion, a perception that these crimes are pervasive. We have related this more clearly to the Ehlers and Clark model. Examples are given to make our explanation clearer and quotes from qualitative studies with analysts are included. Please see p. 17-18.

	what is meant by frequent and in reference to what benchmark.	
R1	4. Line 139 and paragraph - I encourage the authors to consider applying the theory to the analysts own memory of what they are reading/viewing in their work, rather than focusing on the memory report of a third party as the primary stimuli source. Yes, the third party's account of their traumatic experience may influence the level of secondary trauma exposure and the subsequent impact on the analyst, but the analyst's own memory and interpretation of that content is likely key to the subsequent psychological impact that may lead to trauma reactions over time.	On re-visiting the manuscript, we realised we hadn't actually made this point about analysts forming memories from what they're exposed to. This point is now made explicitly on p.17. We've also added information that might not be known by readers about the work of analysts (e.g., volume of crimes worked on in a given period based on personal experience – see p. 16). We have dedicated a specific section of the paper to considering factors associated with the type of material/how it is worked with and how this relates to Ehlers and Clark's, Brewin's and other's models/theories of memory and PTSD (see p. 16 onwards). We hope this now makes the manuscript clearer to read.
R1	5. Line 160 - report the known validity of the SAM and VAM as mechanisms, and acknowledge the criticisms of these components in the PTSD experience.	We have reported Pearson and colleagues' research (p. 9), which challenges some assertions made in Brewin and Ehlers and Clarks models, and suggests that traumatic material might be encoded into autobiographical memory and that conceptual processing might not be protective if it increases the number of cues. This is very relevant to the work of analysts.
R1	6. Where does mentalizing fit within the expanded model? Is it a vulnerability factor in how memory works when exposed to secondary trauma that influences encoding and memory retrieval - or is in in the appraisal per see or a sensory experience issue? I also do not see it represented as a factor in Figure 1.	Mentalising is part of social cognition (as per the original ms), however we have expanded our discussion of mentalising within this section (p. 10) and we return to it when discussing individual differences between analysts (see p. 23). We have explained that it is relevant to the maintenance of PTSD in affecting ability to understand one's mental states and those of others and therefore it can affect help-seeking and sense-making of experiences and associated emotions/cognitions. We also discuss it as part of social support within the workplace (p. 25).
R1	7. When discussing the role of suppression, clarity is required to separate out the effects (or theoretical effects) of suppression during the event itself (or at initial exposure to the problematic stimuli) versus post-event suppression, which relates to recall effects rather than how an event was encoded initially. Presenting the literature to discuss these two phases of	We agree and now separate out discussion of suppression at encoding and suppression at recall (see p. 11 and p. 12).

	when suppression may occur would be helpful in clarifying the findings in the literature and understanding the influence on trauma reactions and persistence over time.	
R1	8. line 337 - "unpick" should be "unpack"	This sentence no longer exists.
R1	9. instead of ending the paper with a list of hypotheses for testing in subsequent research (I would move those to a table), the authors should conclude with potential intervention or prevention strategies that would be helpful for analysts to use based on their theoretical arguments - what is the value of this theoretical argument and expansion of the Ehlers and Clark model to analysts? That should be the take-home message of the paper.	As requested we have moved these to a table (see Table 1). The focus of our ms is to propose a theoretically-informed research agenda for these groups of professionals. We are wary of suggesting interventions when the mechanisms of harm are to be established and such interventions haven't been tested in these settings.
R2	Several questions remain: e.g. what is traumatic material, how does the frequency or dose of exposure count, is there a measure for the extreme traumatic material? Is there information on the duration of the work exposures, in terms of days working? These just some of the variables that also need to be taken into account to expand the cognitive model.	We now define traumatic material on p. 3 with an appropriate reference to leading authors in this field who also use this term. Dosage is given its own section on p. 16. We have added in information from one author's personal experience of being a crime analyst to give a sense of the volume of exposure in 18 months (see p. 16). There isn't published information on how many days a week people work on traumatic material. We do, however, refer a just-published paper which gives a personal insight on working with distressing content (for research) and limiting days working on it (Burrell et al., 2023). As explained above, with our new structure, a whole subsection of the paper focuses on the ways in which analysts work with this type of material and how this relates to Ehlers and Clark's models (from p. 15 onwards). More illustrative examples are given throughout and more quotes are included from published qualitative papers with analysts to bring points to life.
R2	The first line of the outlook paragraph states the central theme of the paper. There should be more focus on this throughout the paper, in terms of evidence, examples, and evidence of the impact, it now fails to give enough leads as to how this can be studied. Based on these limitations, the hypotheses are raising more questions.	With the restructure of the paper, this key point is made at the start of the manuscript but is then returned to with a whole section about the work of analytical staff specifically. There is no quantitative research with these groups that is currently published which is why the paper is mapping out a theoretical framework to guide this future research. However, we have now included rates of STS for other staff exposed to

		secondary trauma making it clear that they tend to work more directly with victims so their exposure is a little different (see p. 15). We have also included figures for number of analytical staff where we could find one reference and we have included a statistic regarding content moderators (see p. 15). As already noted, more illustrative examples of analytical professionals' work are given, as requested.
R2	As authors state repeatedly that longitudinal studies are needed to unpick issues, they do not specify what these studies need to do, answer or have in addition then being longitudinal.	This is now clearer in the manuscript and referred to at relevant points when discussing potential mechanisms and also at the end of the ms.
R2	The definition of what traumatic is in the material needs to be understood before the impact and the context in which this is occurring can be reviewed.	As above, this is now defined. Lots of examples are given regarding the type of work these professionals do and the material they work with which should now make this clearer.
R2	The paper is presented as a review in line 83, but it is not clear what kind of review this paper is providing.	It is a narrative review since its focus would be too wide for a systematic review. However, with the re-write, this sentence is no longer present.
R2	The authors should do better and be talking about 'potential' impact when it comes to trauma. It is not so that the impact is similar for everyone.	We have amended the ms to use "impact" when research has demonstrated a negative impact already or where we are reporting on a theory or hypothesis from another source, and we have used "potential" impact where we are proposing a possible relationship.
R2	The job demands bring on exposure to traumatic material. A discussion would be relevant if it matters that the exposure is anticipated, and not a surprise. The exposure/analysis of elements of the material is part of the assignment.	Predictability is mentioned in Ehlers and Clark (2000) as a protective factor linked to conceptual processing. We have included this now in the paper (p. 7). We return to predictability when we talk about the application of this model to analytical professionals and give examples where unpredictable exposure could still occur as well as including qualitative findings where analysts discuss the benefits of being prepared for exposure (p. 20).
R2	Why is research in the area sorely needed, please motivate once more. The motivation should be based on empirical data about symptom manifestations leading to the inability to perform the job (criterion F, G).	In the previous version of the ms, we had already provided qualitative findings of PTSD/STS symptoms in this population, however there is no published quantitative research on rates for this group. Instead, we have cited rates for other groups of professionals, as already mentioned above. We have also added more quotes from participants in analytical roles to illustrate the impact this work is having. Finally, where we have been able to find figures for numbers of

		professionals in these roles, we have cited them as already mentioned.
R2	Post-traumatic distress is not post-traumatic stress disorder.	Noted. We have been clearer in our use of terminology, making the relationship between PTSD and STS clear and using “PTSD/STS” throughout the paper except where papers are referring to one in particular.
R2	What is the role of circumstantial factors, if someone just becomes a father, or has a death in the family, would they not be more vulnerable to the distress potentially leading to disorder-like experiences/symptoms. Is leadership not aware of this and removes the person from their job temporarily? It is unlikely that ‘one’ exposure leads to PTSD. It is (likely) repeated exposure. So the paper needs also to mention the (repeated) pressure in job situation. It would be interesting to see what impact is of measures to take distance or request (temporary) replacement.	We have referred already to changes in life-circumstances that can make traumatic material more meaningful in terms of threat to an analyst. We have, however, made this section clearer (see p. 8 and p. 24). We have also noted that stress in general can affect mentalising ability. We don’t have scope to consider stress more generally as a vulnerability factor as the paper is already long. We agreed regarding repeated exposure, and we have a whole section now on dosage and how we hypothesise repeated exposure is related to PTSD/STS. Yes, it would be interesting to see what impact such interventions have. From our qualitative research with analysts, some are good at taking breaks and others find this difficult with the pressure of targets. Most do not want to be reassigned and actively avoid disclosing difficulties because of fear of this. We have talked more about taking breaks and managing dosage as an organisation on p. 19 and p. 25.
R2	The paragraph starting with a review of mechanisms is valid; appraisal, is context dependent; threat is ok, unfairness is not well worked out. In some, it may be a motivation to pick a job just in working to fight against the unfairness.	We have added more information about Ehlers and Clark’s model and appraisals specifically and have related these more clearly to the work of analysts with qualitative evidence cited in support (p. 18-19). A section has been written about Belief in a Just World and its potential links with appraisals of unfairness (p. 24) so this should now be clearer.
R2	The prevalence of rapes and other traumatic events is not a valid precedent for claiming that exposure to it is traumatic.	I don’t think we have claimed this but we have tried to make it clear in the manuscript how dosage (as well as exposure to a range of situations in which a crime type occurs) leads to the perception (appraisal) that crime is pervasive. Several studies with analysts have documented these appraisals therefore these are cited to support our argument. It is also made clearer that we are not referring to exposure to the fact that these crimes occur that is traumatic but it is engagement in detail with large volumes of them which is potentially traumatising.

R2	Why emphasize rape and stimulus generalization, when other forms of trauma are also prevalent and may cause more workplace distress or trauma. E.g. forensics in police and crime cases. Here the examples given in the introduction could be more worked out to make a better case.	We have tried to give more examples that are not just related to sexual violence (e.g. a study of forensic science professionals). This is difficult since the vast majority of qualitative research has focused on analysts working on this type of material.
R2	Senses matter – also weak since a lot of material is not only visual and auditory but also reading, or in movie-like presentation. It is to be demonstrated if the reference can be extrapolated. The reference is on learning, wonder if the statement is well referenced with this paper, which is on learning.	We had already indicated in the original manuscript that some analysts were reading the material or watching video footage. We have expanded this section in our revision to make this clearer. We have removed this reference and instead used Ehlers and Clark and other research to explain our hypothesis. We agree that this is to be demonstrated but in this paper we are not presenting evidence for a model but proposing one that can then be the focus of research.
R2	Some reference could be better chosen, e.g. when it comes to early life trauma.	We have revisited all sections to identify any better references. We have included further references for previous life traumas (p. 24).
R2	Extending Ehlers and Clark to Mental imagery – and suppression at encoding and emotion regulation and social cognition. It is to be regretted that they do not carry the focus of the paper on workplace.	This is now addressed with the re-structure of the manuscript. An entire sub-section relates each of these points to the work of analysts.
R2	Sleep is a causal and maintaining factor – this is an example of unclear information. The paragraph lacks focus. Is not linked to the topics of the paper. Half a page is on sleep that is unrelated to the title.	Sleep is included because it is implicated in both the development and maintenance of PTSD. This was stated in the original ms. We have included a new sentence that explicitly makes this point so this is clear to the reader (p. 13). We also include citations to qualitative research which support analysts having difficulties with sleep (p. 23).
R2	Social cognition only has a few lines on social support and support seems very important. How is this weighted in relation to social cognition.	Social support is given greater focus in the revision (see p. 25). It is also referred to in the context of mentalising (an aspect of social cognition).
R2	The figure is interesting but lacks the finetuning that is needed	We have re-visited the figure and amended it particularly given the inclusion of more factors/detail.
R2	Line 335 TPJ and dmPFC , spell out.	These are part of a direct quote, however, we have added a footnote where they are given in full (see p. 11).

3rd Nov 23

Dear Jess,

Thank you for your patience during the editorial evaluation and peer-review process. Your manuscript titled "A Model for Secondary Traumatic Stress Following Work-Place Exposure to Traumatic Material in Analytical Staff" has now been seen by the same 2 reviewers as before, and I include their comments at the end of this message.

The reviewers find your work much improved, but note some remaining concerns regarding structure and clarity of the argument.

We are very interested in the possibility of publishing your Perspective in Communications Psychology, but would like to consider a revised manuscript before we make a decision on publication.

To aid you with that task, I have included a marked-up version of your manuscript.

In sum, we invite you to revise your manuscript taking into account all reviewer comments, guided by the editorial feedback on the manuscript.

EDITORIAL POLICIES AND FORMATTING

You will find a complete list of formatting requirements following this link:
<https://www.nature.com/documents/commsj-style-formatting-checklist-review-perspective.pdf>
Please use the checklist to prepare your manuscript for resubmission.

* TRANSPARENT PEER REVIEW: Communications Psychology uses a transparent peer review system. This means that we publish the editorial decision letters including Reviewers' comments to the authors and the author rebuttal letters online as a supplementary peer review file. However, on author request, confidential information and data can be removed from the published reviewer reports and rebuttal letters prior to publication. If your manuscript has been previously reviewed at another journal, those Reviewers' comments would not form part of the published peer review file.

If you have any questions about any of our policies or formatting, please don't hesitate to contact me.

Please use the following link to submit your revised manuscript (which should be in a separate document to any cover letter):

[link redacted]

We hope to receive your revised paper within 8 weeks; please let us know if you aren't able to submit it within this time so that we can discuss how best to proceed.

Please do not hesitate to contact me if you have any questions or would like to discuss these revisions further. We look forward to seeing the revised manuscript and thank you for the opportunity to review your work.

Best wishes,
Marike

Marike Schiffer, PhD
Chief Editor
Communications Psychology

REVIEWERS' COMMENTS:

Reviewer #2 (Remarks to the Author):

Thank you for the responsive comments to my original review. I have now had an opportunity to review the revised manuscript. The discussion of the literature is better organized and articulated in the revision, including surrounding the operational definitions of the constructs and issues being presented as relevant to analytical staff, thus making the theoretical discussion paper more compelling. I would not call it a review paper per se as referenced in the authors' comments to reviewers; this manuscript is not a systematic analysis of the literature on a topic (given not enough exists), but rather an argument focused on a specific theoretical discussion that uses Ehlers and Clark's model to demonstrate how PTSD/secondary trauma symptoms develop and may be maintained in this unique occupational role of an analysis exposed to difficult content via their work.

1. The Ehlers and Clark model and the additional components argued to expand on this model are sufficiently explained in the revised document, but the links back to the experience of analytical professionals is missing as the reader digests this information and begs the question of what does this have to do with analytical staff? Ultimately, this answer is provided in its own section that elaborates on these points; however, I think the lead in to the application to analytical staff section on page 15 and the first half of page 16 (i.e., prevalence rates and the personalized points and experience of workers) should precede the description of the Ehlers and Clark model so the reader already knows where the authors are going with their elaboration of that model and have the necessary context to think about as they read the authors' description and theoretical expansion. This content would also add to the justification for expanding a model relevant to those who work in analytical roles with secondary trauma exposures. Once the model and expansion is described, the specific breakdown of the model and its added features starting on the latter half of page 16 would be better situated for the reader.

I see value in this theoretical discussion, especially as it applies to an underserved occupational role with trauma vulnerability inherent in the nature of the work.

Grammar points for the entire manuscript:

- whenever "if" is included in a sentence, and "then" is required as well - e.g., If x, y, z, then a, b, c.

- Whenever "this" is used in a sentence, it must be immediately followed by what "this" is referring to in the sentence. e.g., This argument, this point, this fact, this theory, this feeling, this finding. Otherwise, it is unclear as to what "this" is referring in the sentence content. Think "this [what]..." Furthermore, "this" cannot end a sentence as done on Page 7 , first full paragraph, first sentence.

- "Also" cannot be used to start a sentence, and should be "In addition", "Moreover", or "Furthermore". (page 10, second sentence from top of page (and in subsequent sections of the manuscript). In the next sentence in this section, "however" used in the middle of the sentence, connecting two clauses, should be preceded by ";" instead of a comma. This correction is needed in other places in the document as well.

- Page 13, middle paragraph - line 6, change "hasn't" to "has not been" as contractions should be avoided in academic writing. Same thing with "can't" on page 17, middle paragraph, last sentence.

Reviewer #3 (Remarks to the Author):

The authors have responded well to the queries.

A few leftover relatively minor questions:

1. The abstract has been left untouched. It is insufficiently describing what the authors have done to come to the model in this work.
2. The additional mechanisms may leave some questions. How does hearing about trauma from someone else in the triage room contribute? This is not mental imagery, or it may lead to that, but what about listening to a caller on the phone describing a situation? Would that qualify in the model towards PTSD? Has this been covered sufficiently?
3. I fail to read where the extension of the model comes from in the paper.
4. the first review did not cover it, but I did not find that cases were described. To come to the model, it could have helped. Please provide a rationale/justification for not doing this. Maybe it falls outside of the scope of analytical staff.
5. As is now, the paper ends open, not with a firm conclusion, and that piece is lacking to give the paper the status that it is seeking.

5th Jan 24

Dear Jess,

I have now read your revisions of your Perspective "A Model for Secondary Traumatic Stress Following Work-Place Exposure to Traumatic Material in Analytical Staff" and I am delighted to say that we are happy, in principle, to publish it in Communications Psychology under a Creative Commons 'CC BY' open access license.

We are now asking for one final set of revisions. If the revised paper is in Communications Psychology format, in accessible style and of appropriate length, we shall accept it for publication immediately. I have attached an edited version of your manuscript, and ask you to attend to each comment in detail.

EDITORIAL REQUESTS:

* Please review the changes in the attached copy of your manuscript, which has been edited for style, and address the comments and queries I have added. If using Word, please use the 'track changes' feature to make the process of accepting your manuscript more efficient.

* Please check whether your manuscript contains third-party images, such as figures from the literature, stock photos, clip art or commercial satellite and map data. If any of the display items in your manuscript (figures, tables, boxes or movies) include images that are the same as, or are adaptations of, previously published images, please fill in the [Third Party Rights Table](http://www.nature.com/licenceforms/snl/thirdpartyrights-table.doc), and return to us when you submit your revised manuscript. This information will enable us to obtain the necessary rights to re-use such material. If we are unable to obtain the necessary rights to use or adapt any of the material that you wish to use, we will contact you to discuss alternative options.

* Communications Psychology uses a transparent peer review system. On author request, confidential information and data can be removed from the published reviewer reports and rebuttal letters prior to publication. If you are concerned about the release of confidential data, please let us know specifically what information you would like to have removed. Please note that we cannot incorporate redactions for any other reasons.

*If you have not done so already, please alert me to any related manuscripts from your group that are under consideration or in press at other journals, or are being written up for submission to other journals (see www.nature.com/authors/editorial_policies/duplicate.html for details).

FORMATTING GUIDELINES:

Please use the attached checklist to prepare your manuscript for final submission. In the following, I also highlight some issues of particular importance.

** Section Headings

Headings should be no longer than 60 characters (including spaces) and should not use punctuation. Please do not use more than two levels of headings.

**** Figures**

Please remove all figures from the main text and upload them individually, one figure per file. To ensure the swift processing of your paper please provide the highest quality, vector format, versions of your images (.ai, .eps, .psd) where available. Text and labelling should be in a separate layer to enable editing during the production process. If vector files are not available then please supply the figures in whichever format they were compiled in and not saved as flat .jpeg or .TIFF files. If your artwork contains any photographic images, please ensure these are at least 300 dpi.

* Figures should be simple and informative — multi-part figures are best avoided. Boxes should occupy no more than half a page in the PDF (less than 500 words) and may include a figure.

*** References**

References appear as superscript Arabic numerals, in order of mention. The reference list mentions references in the numerical order in which they are mentioned in the main text. If a reference is cited more than once, the same number is used throughout the text and the reference receives a single entry in the reference list.

We ask that you select the most significant 5–10% of references in your list for highlighting, and add a single sentence in bold after each of these references to describe the main result and its significance.

Only papers that have been published or accepted by a named publication should be in the reference list (preprints and citations of datasets are also permitted). Unpublished/Submitted research should not be included in the reference list; it should only be mentioned briefly and parenthetically in the main text. Note that no major arguments should rely on unpublished research.

Published conference abstracts and URLs for web sites should be cited parenthetically in the text, not in the reference list.

Footnotes are not used.

*** Competing interests**

Please include a "Competing interests" statement after the References. Note that we ask authors to declare both financial and non-financial competing interests. For more details, see <https://www.nature.com/authors/policies/competing.html>. If you have no financial or non-financial competing interests, please state so: "The authors declare no competing interests."

SUBMISSION INFORMATION:

In order to accept your paper, we require the following:

* The final version of your text as a Word or TeX/LaTeX file, with any tables prepared using the Table menu in Word or the table environment in TeX/LaTeX and using the 'track changes' feature in Word.

*The completed checklist.

* Production-quality versions of all figures, supplied as separate files. Photographic images should be 300 dpi in RGB format (.jpg, TIFF or native Photoshop format) and any labels/scale bars included in a separate layer from the image. Line art, graphs and schemes should be vector format (.ai, .eps, .pdf); Adobe Illustrator files are preferred and will minimize production time. Any chemical structures or schemes contained within figures should additionally be supplied as separate Chemdraw (.cdx) files.

At acceptance, the corresponding author will be required to complete an Open Access Licence to Publish on behalf of all authors, declare that all required third party permissions have been obtained.

Please note that your paper cannot be sent for typesetting to our production team until we have received this information; **therefore, please ensure that you have this ready when submitting the final version of your manuscript.**

ORCID

Communications Psychology is committed to improving transparency in authorship. As part of our efforts in this direction, we are now requesting that all authors identified as 'corresponding author' create and link their Open Researcher and Contributor Identifier (ORCID) with their account on the Manuscript Tracking System (MTS) prior to acceptance. ORCID helps the scientific community achieve unambiguous attribution of all scholarly contributions. For more information please visit <http://www.springernature.com/orcid>

For all corresponding authors listed on the manuscript, please follow the instructions in the link below to link your ORCID to your account on our MTS before submitting the final version of the manuscript. If you do not yet have an ORCID you will be able to create one in minutes.

IMPORTANT: All authors identified as 'corresponding author' on the manuscript must follow these instructions. Non-corresponding authors do not have to link their ORCID but are encouraged to do so. Please note that it will not be possible to add/modify ORCID at proof. Thus, if they wish to have their ORCID added to the paper they must also follow the above procedure prior to acceptance.

To support ORCID's aims, we only allow a single ORCID identifier to be attached to one account. If you have any issues attaching an ORCID identifier to your MTS account, please contact the [Platform Support Helpdesk](http://platformsupport.nature.com/).

[link redacted]

We hope to hear from you within two weeks; please let us know if the process may take longer.

Best wishes,

Marike

Marike Schiffer, PhD
Chief Editor
Communications Psychology